# An approach to social flexibility: Congruency effects during spontaneous word-by-word interaction

Diana Schwenke[1]*, Tatiana Goregliad Fjaellingsdal[2], Martin G. Bleichner[3], Tobias Grage[1], Stefan Scherbaum[1]

1 Department of Psychology, Technische Universität Dresden, Dresden, Germany, 2 Department of Neurology, University of Lübeck, Lübeck, Germany, 3 Department of Psychology, University of Oldenburg, Oldenburg, Germany

* diana.schwenke@tu-dresden.de

**Data Availability Statement:** All relevant data are within the manuscript and its Supporting Information files. All additional data and data script for analyses are openly available and can be

## Abstract

Cognitive flexibility is the ability to switch between different concepts or to adapt goal-directed behavior in a changing environment. Although, cognitive research on this ability has long been focused on the individual mind, it is becoming increasingly clear that cognitive flexibility plays a central role in our social life. This is particularly evident in turn-taking in verbal conversation, where cognitive flexibility of the individual becomes part of social flexibility in the dyadic interaction. In this work, we introduce a model that reveals different parameters that explain how people flexibly handle unexpected events in verbal conversation. In order to study hypotheses derived from the model, we use a novel experimental approach in which thirty pairs of participants engaged in a word-by-word interaction by taking turns in generating sentences word by word. Similar to well established individual cognitive tasks, participants needed to adapt their behavior in order to respond to their co-actor's last utterance. With our experimental approach we could manipulate the interaction between participants: Either both participants had to construct a sentence with a common target word (congruent condition) or with distinct target words (incongruent condition). We further studied the relation between the interactive Word-by-Word task measures and classical individual-centered, cognitive tasks, namely the Number-Letter task, the Stop-Signal task, and the GoNogo task. In the Word-by-Word task, we found that participants had faster response times in congruent compared to incongruent trials, which replicates the primary findings of standard cognitive tasks measuring cognitive flexibility. Further, we found a significant correlation between the performance in the Word-by-Word task and the Stop-Signal task indicating that participants with a high cognitive flexibility in the Word-by-Word task also showed high inhibition control.

## Introduction

The flexibility to adapt to a human counterpart lies at the heart of human social interaction. This is particularly evident in conversation, where turn-taking happens fluently and both conversation partners respond and adapt flexibly to expected or less expected utterances by the counterpart. The human ability to flexibly adapt to ongoing events has been studied in

downloaded at the open science framework at osf. io/awyfe/.

**Funding:** This project was funded by Volkswagen Foundation grant 89426 (recipient: Stefan Scherbaum) and the Open Access Funding by the Publication Fund of the TU Dresden. The funders had no role in study design, data collection and analysis, decision to publish, or preparation of the manuscript.

**Competing interests:** The authors have declared that no competing interests exist.

individuals for decades in well-established cognitive tasks, such as task switching or set-shifting paradigms [1–3]. Though much has been learned by using those tasks, one might question to which extent such abstract, computer-based experimental studies can capture the complex adaptability of human cognition, especially in social and communicative contexts. To account for this social flexibility, it would be best to address these components in an empirical setting that has elements of social interaction.

Here, we ask which possible mechanisms play a role for social flexibility. To answer this, we propose a model that captures the essential characteristics of social interaction, namely the recurring patterns between two autonomous co-actors [4]. From this model, we derive hypotheses which we then test in a novel social flexibility task where two co-actors engage in an open-ended and co-regulated interactive sequence.

## The importance of co-representation and expectation building

Our ability to continuously adapt our behavior to our environment is an important part in social interaction, where we have to respond quickly and adequately to our partner's behavior. Such behavior is favored by so-called 'co-representations', i.e., a mental representation of our partner's current tasks, behavior and behavioral consequences [5–7]. Such representations have been studied with adapted versions of classical individual cognitive tasks, for example the Simon task where stimulus (e.g., a green or red item) and response (left or right keypress) are either on the same or on opposite sides, which leads to a so-called spatial compatibility effect [8, 9]. In a variant of this task that involved two participants, each individual participant only responded to one stimulus dimension. Both, stimulus and response dimensions were shared by the participants. Interestingly, the original spatial compatibility effect from the individual task still occurred (even though it is not required to take the partner's action into account) [10, 11]. This has been interpreted as evidence that the mere presence of another person automatically induces a mental representation of the other participant's task. While in those cases co-representation compromises the capacity of cognitive resources in the individual mind [12–16], in many other cases it proves to be a major advantage, in particular when acting together towards a joint goal. Here, co-representation implies that we can predict our partner's actions and integrate them into our own behavioral planning [5, 17, 18].

One example of the importance of such expectation building is turn-taking in verbal communication where two people repeatedly alternate between listening and speaking. Surprisingly, the average gap between each turn of around 200ms [19] is much shorter than the average time of 600ms people need to plan their next utterance [20]. Here, co-representation comes into play. It has been suggested that people coordinate their utterances by predicting their interlocutor's upcoming utterance and its end [21–23]. Interestingly, people predict automatically even if they are just reading or listening [24, 25], but depend on having a partner in comparison to acting alone [26]. One can conclude that rapid turn-taking in conversation depends on two factors: First, a valid representation of the ongoing dialog which triggers a fast prediction of future events and, second, planning a fitting response at an early stage [27]. However, this expectation building comes at a cost. While it speeds up the communication as long as expectations are met, it hampers the conversation when the expectations are violated. When the planned response is not valid anymore, it is necessary to abandon the planned utterance and to generate an alternative response that fits the current input.

## Cognitive flexibility

The ability to focus and to switch has been studied by cognitive psychologists under the term cognitive flexibility for decades, e.g., in task-switching or set-shifting paradigms for decades

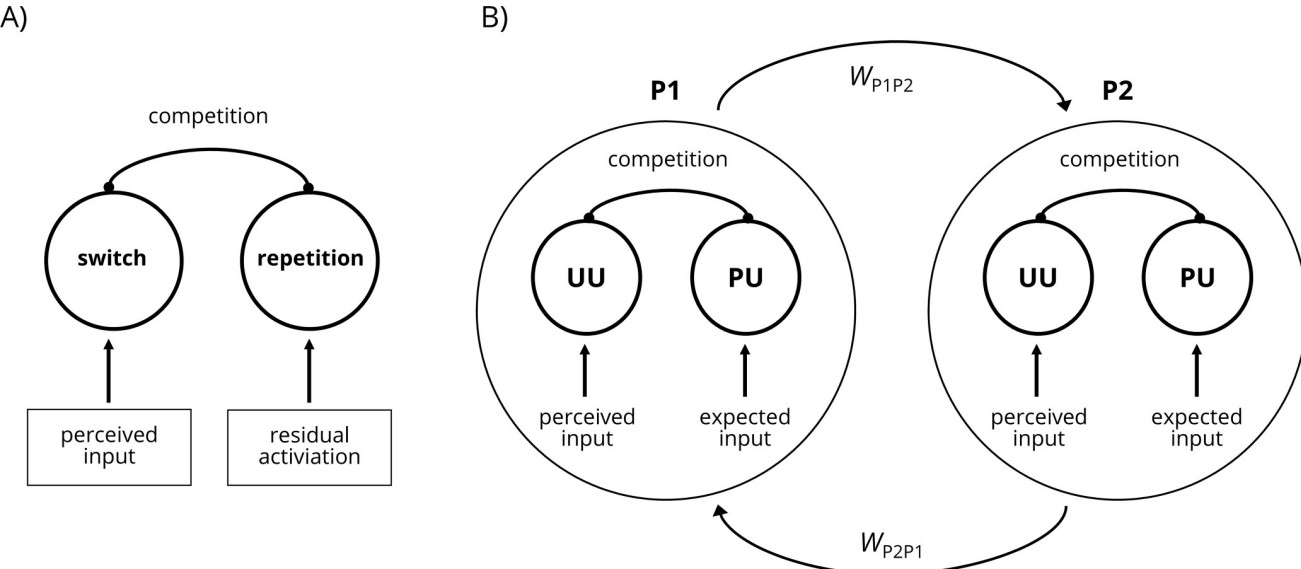

**Fig 1.** Possible mechanisms in A) task-switching and B) social flexibility. A) In critical switch trails a response conflict occurs due to the competing activation of two alternative responses. The switch response is supported by the perceived input but competes with the alternative response that is supported by the residual activation from the previous trial. Additionally, the competition between both responses might suppress the switch response by mutual inhibition. B) In case one co-actor utters an unexpected event, co-actor 2 experiences a response conflict which occurs due to the competing activation of two alternative responses. The unplanned utterance (UU) is supported by the perceived input of the co-actor's last utterance via coupling and the planned utterance (PU) is supported by the expected input as a result of the expected progression of the whole sentence. Additionally, the mutual competition between both units might reduce the activation of the PU which can facilitate a quick and adequate response even if the perceived input is unexpected.

[1–3]. In an exemplary task-switching paradigm, participants have to perform either a magnitude task (is a digit bigger or smaller than 5?) or a parity task (is a digit odd or even?) depending on e.g., the stimuli's color. Trials alternate between 'repetition trials' with the same rule as in the previous trial and 'switch trials' in which the rule changes. By comparing both types of trials, the so-called switch-cost effect then indicates the individual level of cognitive flexibility, i.e., the lower the switch cost the higher the level of flexibility. As a counterpart to switch-costs, compatibility effects as described above for the Simon task, have been interpreted as a measure of how strongly or weakly a person devotes one's resources exclusively to one task-dimension and hence suffer from inflexibility [28, 29].

From a theoretical point of view, different control processes determine how the relevant stimulus is proceeded in order to perform the appropriate response [30–32]. In analogy to the model of the possible mechanisms involved in decision making [32], one can suggest that when facing two alternative responses in the task-switching paradigm (i.e., repetition or switch), both alternative responses are mentally represented and compete with each other (Fig 1A). The pattern of activation will eventually determine which response will be triggered. In repetition trials, no response conflict occurs because the perceived input and the residual activation both activate the same response which quickly triggers the expression of the correct response. In a switch trial, in contrast, the correct response to switch is activated by the perceived input, whereas the alternative false answer to repeat the same response is activated by a residual activation from the previous trial. Additionally, the mutual inhibition (competition) between both options will also reduce the activation of the correct response as long as the incorrect response is still activated. This pattern of activation causes a delayed or even a false response. In summary, the model reveals different mechanisms that explain how unexpected

events in the task-switching paradigm will be processed: the perceived input, the residual activation and the competition between both responses.

If we assume that cognitive flexibility is a necessary process that also underlies social flexibility, e.g., in conversation, this poses the question how we can apply this model and the postulated mechanisms to turn-taking in verbal conversation (especially when unexpected utterances violate already planned responses). In order to answer this, we adapted the original model for cognitive flexibly to the idea of turn-taking in verbal conversation. We then extended the model by a second network analogue to models of co-regulated movement coordination [33]. With this extension we approach social flexibility from an interactive point of view as the model captures the essential characteristic of social interaction, the recurring patterns between two autonomous co-actors. In particular, in this model two coupled networks represent the partners P1 and P2, each of them again with two units representing the competing responses (Fig 1B). The alternative responses here represent the individual tendencies towards the next utterance, i.e., an already planned utterance (PU) based on the expected progression of the whole sentence and an unplanned utterance (UU) that needs to be newly formed. (From a semantic network view, the activation of several more or less expected utterances is possible. For reasons of simplicity, however, the model refers to only *one* expected or unexpected event). Importantly, both networks are coupled so that the UU is activated by the co-actor's network performed utterances (perceived input), while the PU is activated by the expected input. Hence, for the two represented utterances two possibilities exist: First, P1 expresses an utterance that has been expected by P2. This leads to an additional activation of the P2's PU which quickly triggers the expression of her planned utterance as a fitting response. Second, P1 expresses an utterance that has not been expected by P2. This leads to a competing activation of P2's UU which needs to overcome the activation of the opposing PU and only then triggers the expression of an utterance as an alternative solution. Importantly, the mutual inhibition between both units reduces the activation of the PU which can facilitate a quick and adequate response even if the perceived input is unexpected. Additionally, the coupling parameter between both co-actors can vary, e.g., due to possible channel of communication as shown in research on joint action [34, 35]. Hence, this asymmetry in the strength of coupling is implemented as an additional parameter *W*. Similar to the model of cognitive flexibility in the task-switching task, three central mechanisms determine how an unexpected utterance is proceeded in social flexibility: the perceived input and the expected input; the mutual inhibition; and an additional parameter of the co-actors coupling. From this, the following research questions can be derived:

I. Do (more) unexpected events lead to less fluent interaction dynamics, i.e., longer response times in mutual turn-taking?

II. Does a high level of individual inhibition facilitate managing unexpected events?

III. Does the strength of coupling have an effect on how participants handle unexpected events?

## Our research

In order to answer these questions, we took inspiration from improvisational theater games [36, 37]. We adapted a format, called the Word-by-Word game (also known as one word at a time) that lends itself for the study of cognitive flexibility in social communication. In this game, two people jointly act as one single person by taking turns in generating sentences word by word. Each co-actor contributes only one word, followed by the next word by the other co-

actor. It requires a high degree of cognitive flexibility to build meaningful, semantically and grammatically correct sentences: Both co-actors need to align their behaviors in order to react on the co-actor's last utterance.

This task has been studied in a version where the participants were completely free in their choice of wording [38]. However, having no further constraints strongly limits the potential to reliably measure cognitive flexibility. To expand these ideas, we hence aimed to combine the advantages of the different approaches: on the one hand, a high level of co-constructed interaction, and on the other hand, clearly constrained behavior that is necessary for experimental control and analysis. We hence set certain constraints to maintain the level of experimental control of cognitive tasks. In order to differentiate between different conditions, e.g., congruent and incongruent trials, we added target words that had to be integrated in the sentence. The implementation of target words allowed us to experimentally control some of the utterances and manipulate their expectancy. In congruent trials both participants had to include the same target word into the sentence. In incongruent trials, the participants had to include different but semantically related target words (e.g., 'sun' and 'cloud'). This variation of the Word-by-Word game allowed us to study cognitive flexibility in an open-ended, interactive, dynamic setting with sufficient experimental control.

Based on the predictions of the model of social flexibility, we hypothesized (H1) that participants would demonstrate faster response times in congruent compared to incongruent trials. In an additional exploratory analysis, we studied the hypothesis (H2) that participants with higher inhibition would have less difficulties to perform incongruent trials. We studied this by comparing the performance in the Word-by-Word task with measures in classical cognitive flexibility tasks, namely shifting ability in the Number-Letter task [39], and inhibition control in the Stop-Signal task and the GoNogo task [40]. Finally, we study whether the strength of coupling has any effect on the word-by-word performance. Since visibility is known as a key to successful coordination in space and time in joint action research [18, 41, 42], we here manipulated whether participants could perform eye-contact with each other as a potential channel to communicate with one another. We hypothesized (H3) that participant's ability to see each other should lead to a smaller difference between incongruent and congruent trials in the Word-by-Word task compared to when they could not see each other.

## Methods

### Ethics statement

The study was performed in accordance with the guidelines of the Declaration of Helsinki (2008) and of the German Psychological Society. An ethical approval was not required since the study did not involve any risk or discomfort for the participants. All participants were informed about the purpose and the procedure of the study and gave written informed consent prior to the experiment. All data were analyzed anonymously.

### Participants

Sixty students of the Technische Universität Dresden, Germany (48 females, *mean age* = 21.98, *SD* = 2.91) participated in the experiment. Each participant was paired with another co-actor based on their time slot preference yielding thirty pairs (21 female-female, 3 male-male, 6 female-male pairs). The co-actors did not know each other before the experiment. All participants had normal hearing ability and normal or corrected-to-normal vision, they were German-native speakers and had no considerable experience in improvisational theatre. They received 15€ or partial course credit for their participation.

We calculated the sample size with an a priori power analysis under the following assumptions: We expected a medium effect size of (about $d = 0.5$) for the difference between congruent and incongruent trials. Based on power analysis using G*Power [43], we needed a sample size of 27 pairs to detect the effect of interest with an alpha error probability of 5%, and a power of 80%. With an additional 5% of participants to account for experimental loss, our sample size was approximately calculated to N = 30 pairs.

## Procedure

Before the experiment, both participants gave informed consent and reported demographic information. The experiment comprised two parts: A) the Word-by-Word paradigm and B) measures of cognitive abilities.

**A) Word-by-word.** Two participants were seated on opposing sides of a table. Each participant had a computer screen in front of her (they could not see the other's screen). The experimenter had an extra table between both participants with a laptop and mouse, the laptop was connected to the participants' screens. Both participants were equipped with a microphone (Renkforce AVLE1) with an audio pop shield. The Word-by-Word task was programmed in Matlab R2016b. We recorded audio data with the Matlab Psychophysics Toolbox [44, 45]. Before the experiment started, the individual voice-key for each participant was calibrated to find the optimal volume threshold for each individual. The voice-key detected speech onsets and offsets of each participant in the Word-by-Word task. To keep the microphone recording as clean as possible, participants were instructed to utter only the words of the experiment and to avoid filling utterances (e.g., 'ehm', 'eh') and other vocal noises (e.g., laughs, throat clearing). Afterwards, participants were instructed by a standardized tutorial, including a warm-up exercise (reading sentences word by word) and two trial rounds.

## Task description

The participants' task was to construct a short story by taking turns for each word (i.e., word-by-word), acting as one narrator. They were instructed to do this as fast as possible but correctly in terms of grammar and semantics. In principle, they were free in their choice of words. However, they had to include a certain *context word* and a *target word* into their story, leading to a trial construction as follows:

The experimenter stated a name to both co-actors, e.g., 'Sarah', at the beginning of each trial. One of the co-actors had to repeat the name to start the trial. The co-actors took turns in starting the trial. After the name was repeated, the experimenter pressed a key to present a *context word* on each co-actor's screen to set the context in which the following scene would take place, e.g., 'school' (see Fig 2A). Co-actors had to build sentences word-by-word to include the context word correctly (e.g., 'Sarah'–'went'–'to'–'the'—'school'). The context word was identical for both participants in all trials. It was sufficient if one of them said the context word. The occurrence of the context word was marked by a key-press of the experimenter. After the context word was included by one co-actor and the current sentence was finished, the experimenter presented a *target word* on the computer screen of each co-actor. The target word was either the same word for both co-actors (congruent trial) or different but related words (incongruent trial), e.g., 'sun' and 'cloud' (i.e., both words should in principle be appropriate in the given context). The co-actors had to continue their story building word-by-word by including the target word(s). For congruent trials, the trial could be completed as soon as the target word was included by one co-actor. In this case the experimenter marked the utterance with a key-press and thereby triggered a blank screen as a sign to finish the sentence and to complete the

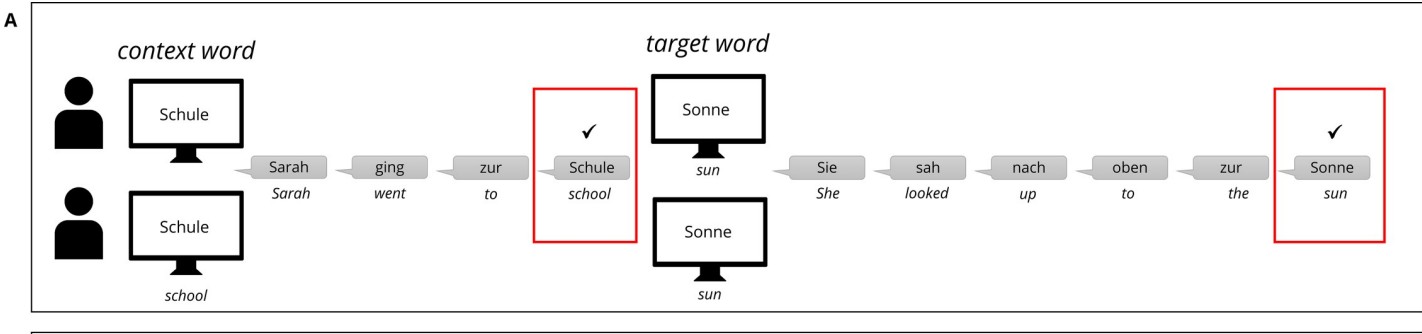

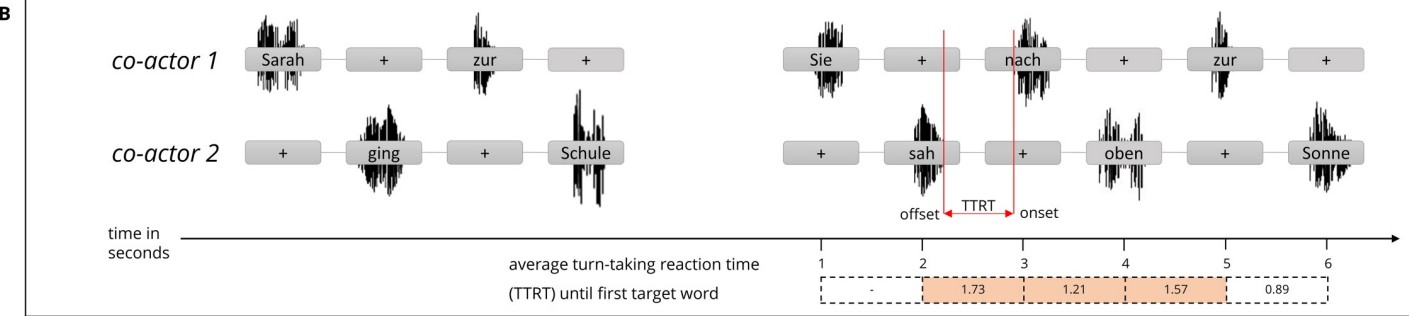

**Fig 2. Word-by-word timeline, example of a congruent trial.** A) A screen in front of each co-actor displayed one context word. Participants engaged in the tasks by contributing only one word, followed by the next word by the other co-actor. When the context word was included, a target word was displayed. After the target word was included, the screen went blank as a signal to end the trial. B) Audio was recorded from both participants concurrently. The turn-taking reaction time (TTRT) was defined as the time between speech offset of one word and the onset of the following word. We only determined TTRTs for the second sentence, i.e., for all transitions between the context and the first target word.

trial. In incongruent trials, both target words had to be included to complete a trial, i.e., the trial continued after the first co-actor had included her target word. The experimenter marked the first target word by a key-press, but did not trigger a blank screen. This was the sign for the first co-actor that the trial was still going on and her co-actor had to include the second target word. After the second target word was included, the experimenter again marked it by a key-press and thereby signalled the end of the trial via a blank screen. Both co-actors finished the sentence to complete the trial.

**Word-by-word material.** For names at the start of each trial, we used 30 female and 30 male names that are common in the German language. Across all pairs, we used the same order of names (male and female names were randomized). We used the same combination of context word and target words for all pairs. This means that we used the same context target combination across all participants, which had to meet three requirements. (1) The context word was always semantically unrelated to both target-words. (2) In the incongruent condition, the target words were always nouns and were always related to each other. (3) In German, there are 3 types of specific articles depending on the item's gender (i.e., 'der', 'die', 'das'). Both target words had the same gender and therefore the same specific article, the same amount of syllables, and they both either referred to a living object (e.g., 'bear', 'teacher') or to a non-living object (e.g., 'book', 'plate'). We tested the context target combinations and the strength of word associations in a pilot study (for a detailed description and results see S1 Data).

**Word-by-word design.** We manipulated two variables, i.e., congruency and visibility. *Congruency*: As a within variable, we manipulated whether the target word was the same for both participants (congruent trials) or only related (incongruent trials). The experiment comprised 60 trials, with 30 congruent and 30 incongruent trials in randomized order. During the

instruction, participants were informed that the target word is either the same or related, but within the experiment they were not aware about the respective condition of an individual trial. *Visibility*: As a between variable, we manipulated whether both co-actors had eye-contact with each other. 15 pairs were seated in front of a partition preventing visual contact. The other 15 pairs had visual contact, but were not instructed to look in each other's eye.

We recorded participants' audio data to measure the turn-taking reaction time (TTRT) for each participant individually (see Fig 2B). TTRT was calculated from word offset to next word onset for each turn-taking. For analyses, we only used TTRTs between the context word and the first target word. We then computed congruency costs by comparing TTRTs in incongruent and congruent trials with lower scores indicating better performance.

**B) Cognitive tasks.** To assess individual differences in cognitive abilities, all participants completed a battery of three cognitive tasks [40]. The participants performed all cognitive tasks at the same time at different working stations with no visual contact. During each of these tasks, participants were allowed pauses of self-chosen duration but were instructed to remain seated. Each task began with a standardized instruction, followed by a short practice block. Tasks were always presented in the same order on a 15-inch monitor.

*Number-Letter.* The screen was divided into four quarters by a horizontal and a vertical line. On each trial, a combination of a letter and a number (e.g., 2a or g9) was presented (3000ms) in the central corner of the quarters. Numbers were either even or odd (2, 4, 6, 8 or 3, 5, 7, 9), letters were either vowels or consonants (a, e, i, u or g, k, m, r). When the stimuli were shown above the horizontal line, participants' task was to indicate whether the number was even or odd. When the stimulus was shown below the horizontal line, participants had to indicate whether the letter was a vowel or a consonant. Key presses were "Y" for even numbers and consonants, "M" for odd numbers and vowels. As the stimulus location was rotated in a clock-wise direction, "switch" trials and "no-switch" trials alternated trial by trial. There were 128 trials in total. We measured the switch costs as the difference between inverse efficiency scores (IES, [40,46]) in switch-trials and IES in no-switch-trials. IES was calculated as the mean RT divided by the frequency of correct responses. Switch costs capture the individualized measure of *shifting ability* (switching flexibly between tasks or mental sets) with lower scores indicating better performance.

*GoNogo.* Each trial started with a fixation cross (750 ms), followed by two dots arranged either vertically or horizontally (500ms). Participants' task was to respond with key-press (space key) when the dots were arranged vertically ("go"), but to withhold a response when the dots were arranged horizontally ("nogo"). The participants had to respond in 280 trials and to withhold in 40 trials. The order was randomized but constrained so that "no-go" trials were separated by at least five "go" trials. We measured the IES as the mean RT in correct go responses divided by the proportion of correct responses in nogo-trials. The IES is an individualized measure of *inhibitory control*, with lower scores indicating better performance.

*Stop-Signal.* Each trial started with a fixation cross (750ms), followed by an arrow pointing to either the left or the right (1000ms). Participants' task was to press the left or the right key ("Y" or "M"), according to the direction of the arrow. In some trials the sideward-pointing arrow was replaced by an upward-pointing arrow after a variable stop-signal delay (SSD). In this case, participants had to withhold their response. The initial SSD of 200ms was adapted after each "stop" trial by adding 50ms for a correct non-response and subtracting 50ms for a false response. This was to achieve a stop-trial error-rate of approximately 50%. Participants had to respond in 200 trials and to withhold in 40 trials. The order was randomized but constrained so that "no-go" trials were separated by at least five "go" trials. We measured the stop-signal reaction time (SSRT) according to the quantile method [47]. As it has been suggested to exclude participants with very low SSRTs [40, 47], we corrected all SSRTs that were lower than

0 to 0. The SSRT is an individualized measure of *inhibitory control*, with lower scores indicating better performance.

With these three tasks, we hope to capture a wide range of different sub-processes of cognitive flexibility [48] as all three paradigms feature particular task requirements: In the Number-Letter task participants perform an ongoing response but have to switch between alternative actions. In the Stop-Signal task participants prepare responses in all trials but have to withhold this response in some trials, whereas in the GoNogo task participants only respond in particular trials.

## Results

We determined participants' average turn-taking reaction time (TTRT) and congruency effects (CE) for the Word-by-Word task as described above. On average, both participants needed more turns to include the target word in incongruent trials $M = 8.30$ ($SD = 2.61$) than in congruent trials $M = 6.24$ ($SD = 1.51$), $t(29) = 5.73$, $p < 0.001$, $d = 1.06$. For the cognitive measures, we calculated switch costs as a measure for cognitive flexibility in the Number-Letter task, inverse efficiency scores (IES) as a measure for inhibitory control in the GoNogo task and stop-signal reaction time (SSRT) as a measure for inhibitory control in the Stop-Signal task as described above. In the following, we only use the task-names for the cognitive measures for easier comprehension.

We first tested whether participants had more difficulties to build sentences in the incongruent condition compared to the congruent condition (H1) and whether this effect was modulated by the participants' ability to see each other (H3). We therefore performed a repeated-measure analysis of variance (ANOVA) on the TTRT with the within-factor *congruency* and the between-factor *visibility*. We found a main effect for *congruency*, $F(1,58) = 63.90$, $p < .001$, $\eta_p^2 = 0.524$, yielding shorter mean TTRTs for congruent trials ($M = 1.74$ s, $SE = 0.067$ s) compared to incongruent trials ($M = 1.99$ s, $SE = 0.062$ s), no main effect for *visibility*, $F(1,58) = 1.45$, $p = .23$, and no interaction effect between both variables $F(1,58) = 1.42$ $p = .240$ (see Fig 3 left). These results indicate that participants had more difficulties to build sentences in trials in which the target word was different compared to trials in which the target word was the same, but this effect was not modulated by participants' visibility.

Next, we tested whether there were significant relationships between the performance in the Word-by-Word task and measures from cognitive tasks (H2). We calculated Pearson's correlations between the Word-by-Word measures (TTRT, CE) and the cognitive tasks' measures (see Table 1). In summary, we found a significant correlation between the congruency cost in the Word-by-Word task and the SSRTs in the Stop-Signal task, indicating that participants who showed smaller differences between congruent and incongruent trials in the Word-by-Word task also showed smaller stop signal reaction times in the Stop-Signal tasks (note that the correlation between CE and non-corrected SSRTs also reached significance with $r = 0.301$, $p = 0.019$). We further found an expectable correlation [40] between the Stop-Signal task and the GoNogo task (see Fig 3 right). No other correlation reached significance.

## Discussion

The ability to flexibly adapt to another person is essential in human social interaction where people have to respond quickly and adequately to our partner's behavior, e.g., in turn-taking during verbal conversation. Despite the fact that flexibility plays a crucial part in our social life, cognitive research has mainly focused on the individual mind for the past decades. The ability to switch between different concepts or to adapt goal-directed behavior in a changing environment has been studied in abstract, computer-based tasks [1–3]. This study aimed to investigate

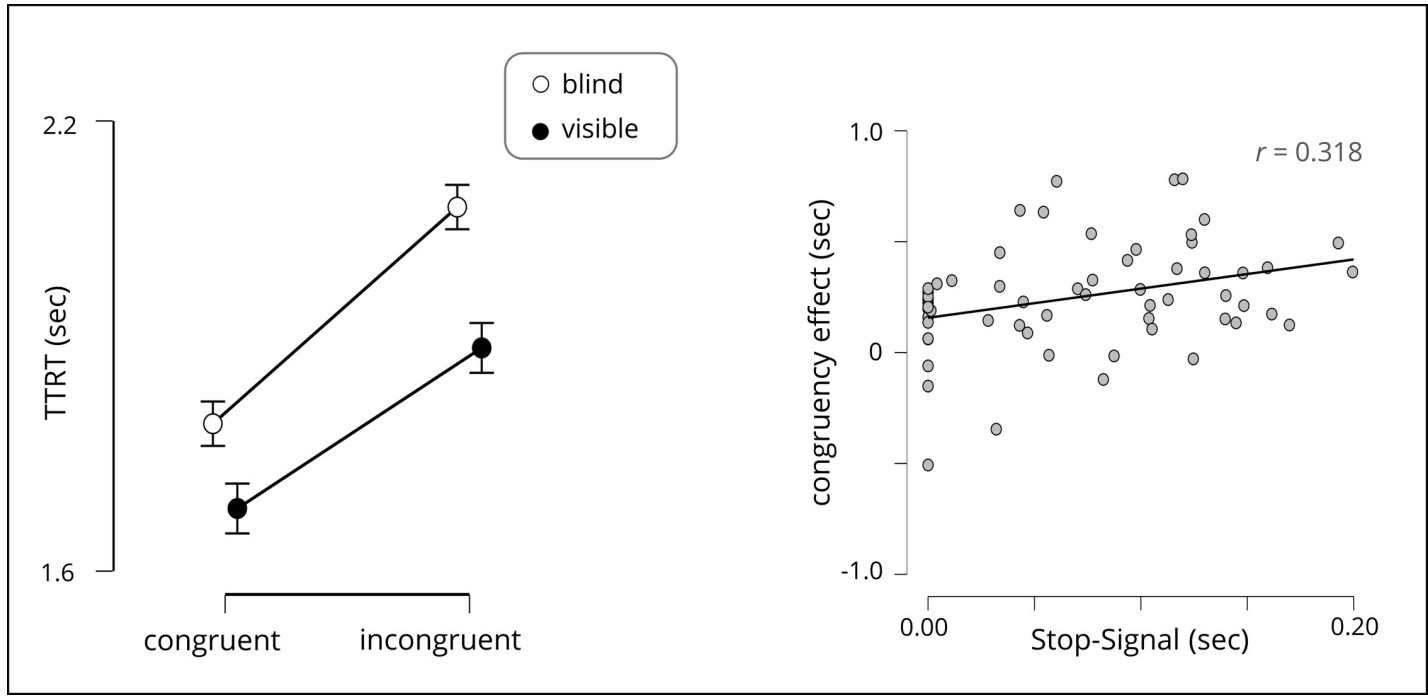

**Fig 3. Results.** Left) Average turn taking reaction time (TTRT) in seconds depending on congruency (congruent, incongruent) and visibility (visible, blind) shown. Error bars indicate standard error over the mean. Right) Pearson correlation between congruency cost in seconds in the Word-by-Word tasks and SSRTs in seconds in the Stop-Signal task.

social flexibility in a turn-taking word-by-word-paradigm and thereby go beyond the study of cognitive flexibility in classical individual tasks. Therefore, we, first introduced a model of social flexibility that captured the essential characteristics of social interaction and especially of turn-taking situations: the recurring patterns between two autonomous co-actors [4]. Based on that model, we derived essential mechanisms that eventually determine how an event is processed: the perceived input and the expected input, the mutual inhibition, and the strength of coupling. In order to investigate these components, we derived hypotheses that we systematically tested in a novel task in which two participants together built sentences by taking turns word-by-word. Participants here had to include target words that were either the same for both participants (congruent condition) or different but semantically related (incongruent

**Table 1. Pearson correlation between all measures.**

|  |  | TTRT | CE | Number-Letter | GoNogo | Stop-Signal |
|---|---|---|---|---|---|---|
| **TTRT** | r | - |  |  |  |  |
|  | p | - |  |  |  |  |
| **Congruency Cost** | r | -0.103 | - |  |  |  |
|  | p | 0.434 | - |  |  |  |
| **Number-Letter** | r | -0.022 | -0.151 | - |  |  |
|  | p | 0.868 | 0.249 | - |  |  |
| **GoNogo** | r | 0.053 | 0.130 | 0.183 | - |  |
|  | p | 0.690 | 0.324 | 0.161 | - |  |
| **Stop-Signal** | r | 0.059 | 0.318 | 0.057 | 0.430 | - |
|  | p | 0.653 | 0.013 | 0.665 | < .001 | - |

condition). In the following, we will go through the findings of our study and discuss them in reference to the model of social flexibility.

## Main findings

We first tested whether unexpected events would induce delayed responses. As predicted, we found a significant congruency effect showing that participants were faster in including the same target word compared to different but related target words. More specifically, on the level of each individual utterance, participants were faster in coming up with the next word in congruent compared to incongruent conditions. This finding confirmed the first assumption of the model, i.e., that the perceived input and the expected input activates competing responses, namely a planned utterance (PU) and an unplanned utterance (UU). In particular, in congruent trials the perceived and the expected input both led to a high activation of the PU which quickly triggered a fitting response. In contrast, incongruent trials led to a competing activation of both units which induced delayed responses. This pattern also reflects the primary outcome of standard cognitive tasks measuring cognitive flexibility, for example in flanker-tasks [49] or homonym judgement tasks [50], where a small congruency cost suggests a high level of cognitive flexibility.

Second, we tested whether the inhibition between both units would lead to a reduced activation of the PP which, according to the model, should facilitate handling unexpected events. We therefore studied the relation between Word-by-Word measures and more classical cognitive measurements, namely shifting ability via the Number-Letter task [39] and inhibition control via the Stop-Signal task and the GoNogo task [40]. We found a significant correlation between the Word-by-Word task and the Stop-Signal task indicating that participants with a high social flexibility in the Word-by-Word task showed also high inhibition control in a classical cognitive task. This leads to the conclusion that participants who have the ability to deliberately suppress dominant, inappropriate, or automatic responses in the Stop-Signal task might find it easier to adapt to an unexpected event which confirms another parameter of the model. However, this finding should be interpreted with caution since the correlation between the Word-by-Word task and the Stop-Signal tasks was only weak. Further, we found no correlation with the GoNogo task, also measuring inhibitory control, and the Number-Letter task measuring task shifting ability. Both non-significant findings may be due to the fact that different cognitive sub-processes support cognitive flexibility which are conceived as partially separate components of our cognitive system [48]. For example, in situations where two conflicted goals require inhibition control, two sub-processes are required to overcome automatic responses: monitoring the environment for potentially new information on the one hand *and* shifting between tasks to manage multiple goals simultaneously on the other hand. In this sense, it is reasonable, that the Word-by-Word task requires inhibitory control but not shifting ability.

Finally, we studied whether the varied coupling between both partners would influence how they handle unexpected events. We found no significant effect for visibility indicating that whether or not participants had visual contact did not influence task performance. Pairs in the group with visual contact had the opportunity to make eye-contact with her co-actor over the screen. Although, since we have not instructed them to do so nor did we determine whether or not they engaged in any eye-contact, it is possible that the pairs did not benefit from this way of communication because they did not use it. A further explanation, however, is that the study was underpowered because our sample size analysis was based on a within-subject design. It is therefore reasonable that the sample size we used in this study was too small for a between-subject comparison. Since research on joint action has shown that the

success of interaction depends on the visibility of the interaction partner [34, 35], a fine-grained investigation of the coupling could be an exciting direction for future research.

## A critical evaluation of the word-by-word task

In this study, we introduced a novel task in order to account for the interactive aspects of human cognitive flexibility. Participants' task was to build meaningful sentences in terms of grammar and semantics word-by-word, which required a high level of flexibility since both co-actors needed to adapt their behavior in order to react to the co-actor's last utterance. Our task was inspired by a game commonly used in improvisational theater [36, 37] that lends itself for the study of social flexibility in verbal turn-taking [51]. It has the capacity to lead to fully free and spontaneous story-telling and thereby models various aspects of natural interactions. However, having no further constraints strongly limits the potential to reliably measure cognitive flexibility. We hence set certain constraints in order to maintain the level of experimental control of cognitive tasks. With this adaptation we combined the advantages of the different approaches: on the one hand a high level of co-constructed interaction, and on the other hand, clearly constrained behavior that is necessary for experimental control, and analysis.

However, due to our aim to measure the cognitive flexibility in social communication, our experimental paradigm critically differs from classic cognitive tasks such as the Simon task or task-switching paradigms [1–3, 8, 9]. In contrast to a visual presentation of imperative stimuli, participants here produced their own material verbally and thereby their own stimuli for their co-actors. This procedure gives the opportunity to measure social interaction in a way that incorporates many aspects of real-life interactions: Both co-actors speak *and* listen reflecting a constant alternation of give and take; they both have to coordinate with one another in taking turns, and the interaction is spontaneous and open-ended. With this implementation, the Word-by-Word task differs in the level of processing from more well-established cognitive tasks since it requires a variety of different higher-cognitive processes such as word-recognition or memory components that we did not address in our theoretical considerations. However, we argue that this is not fundamentally different in comparison to classical cognitive tasks. Firstly, 'simple' cognitive tasks that measure cognitive flexibility also include a variety of different sub-processes that are necessary to execute the task–but not targeted by the research question they aim to answer. Secondly, this is why we (and cognitive psychologists usually) perform within-subjects measures that use the same task with the same set of many processes involved across conditions and varied properties between conditions that should influence the processes of interest.

Nonetheless, the level of real-life interaction might be "too" natural for some research disciplines. When interested in the neural underpinnings, for example, a much higher degree of experimental control is needed. This is mainly due to the measurement methods being prone to noise and the need for averaging over multiple, at best identical, experimental trials [52]. Despite these challenges, the interest in moving toward a neuroscience of social interaction is high [53]. The word-by-word technique used in this study involves multiple aspects that can be targeted from a neuroscientific perspective. One obvious aspect is the verbal character and subsequently the language processing coming into play when forming sentences word-by-word. Aspects that we have targeted in our former studies, where we have used more controlled versions of the Word-by-Word game and found significant N400 and P600 ERP effects when the participant's expectation was deliberately violated by a co-acting confederate [54]. We further found amplitude differences for the P200, N400, and P600 ERPs when participants had to handle unexpected events in a scripted word-by-word interaction [55]. These results clearly indicate that the Word-by-Word task can serve as a reliable tool for measuring

behavioral as well as neuronal aspects of social flexibility. We can hence conclude that this task serves as a valid tool for an investigation of cognitive *and* neuronal processes that also include interactive aspects of human cognition. Future studies can build on our findings and further our understanding on how expectancy effects and the preparation, inhibition, and production of alternatives model human interactions.

## Summary

In view of existing research on human cognition, we contribute a novel and innovative task that measures cognitive flexibility in social interaction. Our results indicate that we successfully managed to replicate a primary outcome of standard cognitive tasks, the effect of larger reaction times in incongruent compared to congruent trials. Toward the goal to capture cognitive flexibility in an open-ended, reciprocal, and dynamic setting, we combined the advantage of a high level interaction on the one hand, and experimental control on the other hand.

## Supporting information

**S1 Table. Pearson correlation between all measures.** *significance at $p < 0.05$ **significance at $p < 0.001$.
(DOCX)

**S1 Data. Rating of the subjective relation between Context Word (CW), Target Word 1 (TW1), and Target Word 2 (TW2).**
(DOCX)

**S2 Data. Minimal data set.**
(CSV)

## Acknowledgments

We thank Judith Herbers for help during data collection.

## Author Contributions

**Conceptualization:** Diana Schwenke, Tatiana Goregliad Fjaellingsdal, Martin G. Bleichner, Stefan Scherbaum.

**Data curation:** Diana Schwenke.

**Formal analysis:** Diana Schwenke.

**Funding acquisition:** Martin G. Bleichner, Stefan Scherbaum.

**Methodology:** Diana Schwenke, Tatiana Goregliad Fjaellingsdal, Stefan Scherbaum.

**Project administration:** Stefan Scherbaum.

**Resources:** Stefan Scherbaum.

**Software:** Stefan Scherbaum.

**Supervision:** Martin G. Bleichner, Stefan Scherbaum.

**Validation:** Tatiana Goregliad Fjaellingsdal, Martin G. Bleichner.

**Visualization:** Diana Schwenke.

**Writing – original draft:** Diana Schwenke.

**Writing – review & editing:** Tatiana Goregliad Fjaellingsdal, Martin G. Bleichner, Tobias
Grage, Stefan Scherbaum.

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
