## [Decision Letter · Decision Letter 0]

7 May 2020

PONE-D-20-00636

A social approach to cognitive flexibility: Congruency effects during spontaneous word-by-word interaction

PLOS ONE

Dear Mrs. Schwenke,

Thank you for submitting your manuscript to PLOS ONE. After careful consideration, we feel that it has merit but does not meet PLOS ONE’s publication criteria as it currently stands. Therefore, we invite you to submit a revised version (major revisions) of the manuscript that addresses the points raised during the review process.

We would appreciate receiving your revised manuscript by Jun 21 2020 11:59PM. To enhance the reproducibility of your results, we recommend that if applicable you deposit your laboratory protocols in protocols.io, where a protocol can be assigned its own identifier (DOI) such that it can be cited independently in the future. For instructions see: http://journals.plos.org/plosone/s/submission-guidelines#loc-laboratory-protocols

We look forward to receiving your revised manuscript.

Kind regards,

Marte Otten, Ph.D.

Academic Editor

PLOS ONE

Journal Requirements:

Additional Editor Comments (if provided):

The two reviewers raise very similar points: the central example that positions your research is confusing, the research question is at best lacking in detail (but, as reviewer 2 indicates, it also does not seem to clearly follow from the literature you present, and the research question that is (implicitly) presented seems to be of little scientific relevance, at least in the way it is communicated now), and the results need to be more rigorously analysed and interpreted.

These issues deal with the backbone of this manuscript, and should therefore be fundamentally addressed before I can consider this manuscript for publication.

Reviewers' comments:

Reviewer's Responses to Questions

**Comments to the Author**

1. Is the manuscript technically sound, and do the data support the conclusions?

Reviewer #1: Yes

Reviewer #2: No

2. Has the statistical analysis been performed appropriately and rigorously? 

Reviewer #1: Yes

Reviewer #2: No

3. Have the authors made all data underlying the findings in their manuscript fully available?

Reviewer #1: Yes

Reviewer #2: Yes

4. Is the manuscript presented in an intelligible fashion and written in standard English?

Reviewer #1: Yes

Reviewer #2: No

5. Review Comments to the Author

Reviewer #1: This paper reports a novel study on participants' ability to create sentences together, providing a new route to test cognitive flexibility in social context. I like the idea for the study and find the results interesting. However, the authors should try to better embed the study in the existing literature, motivate the predictions more clearly, report an additional analysis on the number of turns taken, and discuss the results in more detail.

Specific comments:

The example of the two friends cooking is nice but has some misleading elements. The fact that they were debating whether to order pizza or to cook spaghetti seems irrelevant. Also, waiting for 30 minutes for a friend to arrive does not seem all that odd. The authors would like to illustrate " the sticking to initial assumptions despite evidence for the contrary ", but is the friend's not being on time clear evidence to the contrary in the example? It would help to clear and sharpen this example case.

The first paragraph is very long and should be divided into two or three.

In the review of relevant literature, it may be useful to consider this study on joint task switching: Dudarev, V., & Hassin, R. R. (2016). Social task switching: On the automatic social engagement of executive functions. Cognition, 146, 223-228.

The authors should also consider including earlier work on joint task performance that has used sentence completion, in particular research by Gami and colleagues:

Corps, R. E., Gambi, C., & Pickering, M. J. (2018). Coordinating utterances during turn-taking: The role of prediction, response preparation, and articulation. Discourse Processes, 55(2), 230-240.

Corps, R. E., Pickering, M. J., & Gambi, C. (2019). Predicting turn-ends in discourse context. Language, Cognition and Neuroscience, 34(5), 615-627.

Furthermore, there is work on joint word production that seems relevant:

Kuhlen, A. K., & Rahman, R. A. (2017). Having a task partner affects lexical retrieval: Spoken word production in shared task settings. Cognition, 166, 94-106.

The hypotheses need more justification. What exactly motivates the prediction that participants should have faster response times on congruent trials? In what sense are incongruent trials truly incongruent, given that the target words to be used by each participant are semantically related? Why do the authors predict that being able to see each other would reduce the effect of congruency? What motivates predictions concerning specific correlations between solo cognitive tasks and joint performance? A clear link between what is known based on the existing literature and the predictions is needed.

Solo Cognitive Tasks: Were pairs of participants tested at the same time? Did they complete the tasks facing each other? This should be specified in the methods section.

Analyses: It would be informative to know the number of turns participants needed to complete the task. Were there more turns in the incongruent condition?

The General Discussion should be expanded to discuss potential mechanisms underlying the observed effects as well as limitations of the study in more detail.

Line 107: delete the question mark

Line 116: co-actors should be co-actor's

Line 141 incomplete sentence

Line 146, Declaration of Helsinki: The authors should state the year, because to comply with the most recent version of the Declaration, pre-registration is mandatory.

Reviewer #2: Summary:

In the paper "A social approach to cognitive flexibility: Congruency effects during spontaneous word-by-word interaction" the authors explore a novel task to measure cognitive flexibility in a more social context than other standard measures of cognitive flexibility.

While the idea of this task is, as far as I can tell, indeed novel, this paper is not suitable for publication in the psychological or psycholinguistic literature, and in my view even less for publication in a general journal like PLOS-ONE.

First, the writing would need to be substantially improved. There are many oddly placed commas (in English, the words "although" and "that", when starting a relative clause, are not followed by a comma), and marked formulations, and a sentence that ends in mid-air (line 141). These errors are of course easily corrected, but my point is that they *should* be corrected before the paper is submitted to a scientific journal, e.g. by asking a native speaker of English, or a colleague with more experience in writing in English, to proofread it. More importantly, many parts of the manuscript, including the abstract, are hard to follow. The example at the beginning of the introduction (two friends who both believe a dinner is taking place at their place) appears to have little to do with cognitive flexibility, but rather with mismatching representations of an earlier-made agreement. The authors write about this: "it often seems striking how the assumptions held by both sides overshadowed the inconsistencies in the initial planning." The reader is left to wonder: yes, how could they not have noticed this misunderstanding, as this is not explained in the text. But whatever the reason, it seems unlikely to have been caused by a lack of cognitive flexibility, because that's not how misunderstandings of this sort typically arise. So, the reader then wonders what the point is of the example and how it relates to the issue of cognitive flexibility. Is it that the friends (or at least one of them) now have to stop cooking and attend to different cognitive tasks?

Second: There is no research question. Normally, the literature is reviewed, an issue arising from this review is presented, and this issue is then condensed into a scientific question of importance (at least for the field), and then addressed with an experimental study aimed at answering that specific question as well as possible. The implicit research question here seems to be "does this new task we invented, which we think might measure cognitive flexibility, and which we have reason to believe is more social and natural than the traditional ones, elicit the effects which we think it should?" This is a very specific and ad-hoc question, and the authors do not make clear how this specific issue is of general interest for the field of cognitive psychology.

Third, there is no good theoretical motivation for the explored task. The authors claim that it is more natural and more social, because it involves one person's utterance to be the input for the next person's task, and so on. This of course creates interdependency between what the two participants do, but that does not necessarily make the task natural or social. The ecological validity of the task is, I believe, very low, and if the authors would want to claim that it is sufficiently high, they need to provide clear arguments for that claim. Personally, I find it hard to imagine many situations situation in our daily lives where the ability to collectively formulate a grammatical sentence by alternatingly and spontaneously providing a word would be useful. Also, the task seems fundamentally different in the level of processing from the three other tasks that it is compared with. Contrary to the other tasks, which all involve cognitive control at relatively peripheral levels, the new task involves a number of simultaneous higher-cognitive subtasks: the parsing of the words, the constructing of higher-level linguistic representations, word recognition, semantic processing, and memory components. Each of these sub-tasks literally have entire scientific sub-fields devoted to studying them. And these sub-processes also run in parallel in the participant's brain, which adds an entire additional level of complexity. This means that there are so many things happening in the participant's mind/brain during this task that whatever the results that were to come out of it, they are very difficult to be pinned down to specific cognitive variables. And I see no compelling reason to believe that the task is more sensitive to cognitive flexibility than to any of the other variables that are involved in the list of simultaneously running sub-processes in my, almost certainly incomplete, list above.

Finally, the results of the study are not encouraging and hard to interpret. First, there was no significant effect on visibility. The authors suggest that that was because the participants may not have chosen to exploit the visual channel, but another possibility is that the study was just severely underpowered, as the power analysis on which the design was based was on a within-participant effect of medium strength. Not on a potentially small effect in a between-participant effect (which has much lower power). Second, the TTRT was not correlated with any of the standard flexibility tasks. Third, the CE difference was only weakly correlated with one of the tasks, while the p-value of that test was not corrected for the number of correlation tests (e.g. using the Bonferroni method). So according to the results, the task seems not to behave as expected or desired. This is, however, not the conclusion that the authors draw.

For these reasons, I believe that this study is not publishable in PLOS-ONE.

6. PLOS authors have the option to publish the peer review history of their article (what does this mean?). If published, this will include your full peer review and any attached files.

Reviewer #1: No

Reviewer #2: No

---

## [Author Response · Author response to Decision Letter 0]

4 Jun 2020

Dear Dr. Marte Otten, 

Thank you and the reviewers for your valuable comments on the previous version of our manuscript PONE-D-20-00636 entitled “A social approach to cognitive flexibility: Congruency effects during spontaneous word-by-word interaction” (now titled “An approach to social flexibility: Congruency effects during spontaneous word-by-word interaction”) which my colleagues and I recently submitted to PLOS ONE. We are happy to submit this major revision addressing all the points raised in the first round of reviews. Further, we uploaded our study’s underlying data set as a supporting information file (minimal data) in addition to our upload of our primary data and analysis scripts which is publicly available at the open science framework osf.io/gufr6/.

In the new version of our manuscript, we added several passages on the theoretical foundation in order to constitute our research question, as proposed by both reviewers. We now present a model that captures the possible mechanisms of social flexibility in verbal word-by-word interaction and thereby provides the essential theoretical foundation to predict specific hypotheses. 

In this rebuttal letter, I will go through all the points in the order in which they were raised in the reviews. The responses are printed in blue. In addition to this letter, you will find all changes highlighted in the reworked version of the manuscript by using colored text. 

Reviewer 1

1. The example of the two friends cooking is nice but has some misleading elements. The fact that they were debating whether to order pizza or to cook spaghetti seems irrelevant. Also, waiting for 30 minutes for a friend to arrive does not seem all that odd. The authors would like to illustrate " the sticking to initial assumptions despite evidence for the contrary ", but is the friend's not being on time clear evidence to the contrary in the example? It would help to clear and sharpen this example case

We agree with the reviewer that the introduction example did not catch the essential aspects of our research. We therefore deleted the example and chose a more general introduction. 

2. The first paragraph is very long and should be divided into two or three.

We thank the reviewer for this advice and offer a new structure in our revised version of the manuscript. 

3. In the review of relevant literature, it may be useful to consider this study on joint task switching: Dudarev, V., & Hassin, R. R. (2016). Social task switching: On the automatic social engagement of executive functions. Cognition, 146, 223-228. The authors should also consider including earlier work on joint task performance that has used sentence completion, in particular research by Gami and colleagues:

Corps, R. E., Gambi, C., & Pickering, M. J. (2018). Coordinating utterances during turn-taking: The role of prediction, response preparation, and articulation. Discourse Processes, 55(2), 230-240. Corps, R. E., Pickering, M. J., & Gambi, C. (2019). Predicting turn-ends in discourse context. Language, Cognition and Neuroscience, 34(5), 615-627. 

Furthermore, there is work on joint word production that seems relevant:

Kuhlen, A. K., & Rahman, R. A. (2017). Having a task partner affects lexical retrieval: Spoken word production in shared task settings. Cognition, 166, 94-106.

We thank the reviewer for these suggestions. We added a new paragraph in which we considered the role of shared task paradigms and language production in direct relation to our research question (page 3-4). 

4. The hypotheses need more justification. What exactly motivates the prediction that participants should have faster response times on congruent trials? In what sense are incongruent trials truly incongruent, given that the target words to be used by each participant are semantically related? Why do the authors predict that being able to see each other would reduce the effect of congruency? What motivates predictions concerning specific correlations between solo cognitive tasks and joint performance? A clear link between what is known based on the existing literature and the predictions is needed.

We thank the reviewer for this request of clarification. The theoretical foundation of the hypotheses we claimed to verify clearly needed a more thorough consideration. We now present a model of social flexibility based on the considerations of Hommel et. al (2011) and Noy et. al (2011) see page 5-7. In this model we describe possible mechanisms that are involved in social flexibility. From these mechanisms we now derive specific hypotheses in terms of the congruency cost, the link between the word-by-word performance and the varied eye-contact between both participants (page 7-8).

5. Solo Cognitive Tasks: Were pairs of participants tested at the same time? Did they complete the tasks facing each other? This should be specified in the methods section.

We thank the reviewer for pointing us towards this lack of clarity. The participants indeed performed the cognitive tasks at the same time. Therefore, they were seated at separate working stations with no visual contact. We clarified this information in the methods section (page 14). 

6. Analyses: It would be informative to know the number of turns participants needed to complete the task. Were there more turns in the incongruent condition?

We thank the reviewer for this important note. Participants in the congruent condition needed on average 6.24 (SD = 1.51) turns until they included the target word. Participants in the incongruent condition needed on average 8.30 (SD = 2.61) turns. Participants needed more turns to include the target word in incongruent trials than in congruent trials, t(29) = 5.73, p < 0.001, d = 1.06. 

We added this data in the results section (page 16). 

7. The General Discussion should be expanded to discuss potential mechanisms underlying the observed effects as well as limitations of the study in more detail.

We agree with the reviewer that our discussion section failed to address potential mechanisms underlying our findings. We hope that with the introduction of our model of social flexibility, these links become more precise. We now structured the main paragraph in our discussion section along those mechanisms in order to address our findings in the context of our theoretical foundation. 

8. Further points: 

Line 107: delete the question mark

Line 116: co-actors should be co-actor's

Line 141 incomplete sentence

Line 146, Declaration of Helsinki: The authors should state the year, because to comply with the most recent version of the Declaration, pre-registration is mandatory.

We thank the reviewer for pointing to these mistakes and changed the sentences accordingly. 

Reviewer 2

1. First, the writing would need to be substantially improved. There are many oddly placed commas (in English, the words "although" and "that", when starting a relative clause, are not followed by a comma), and marked formulations, and a sentence that ends in mid-air (line 141). These errors are of course easily corrected, but my point is that they *should* be corrected before the paper is submitted to a scientific journal, e.g. by asking a native speaker of English, or a colleague with more experience in writing in English, to proofread it. More importantly, many parts of the manuscript, including the abstract, are hard to follow. The example at the beginning of the introduction (two friends who both believe a dinner is taking place at their place) appears to have little to do with cognitive flexibility, but rather with mismatching representations of an earlier-made agreement. The authors write about this: "it often seems striking how the assumptions held by both sides overshadowed the inconsistencies in the initial planning." The reader is left to wonder: yes, how could they not have noticed this misunderstanding, as this is not explained in the text. But whatever the reason, it seems unlikely to have been caused by a lack of cognitive flexibility, because that's not how misunderstandings of this sort typically arise. So, the reader then wonders what the point is of the example and how it relates to the issue of cognitive flexibility. Is it that the friends (or at least one of them) now have to stop cooking and attend to different cognitive tasks?

We understand that the introduction example did not catch the essential aspects of our research. We therefore deleted the example and chose a more general introduction. Further, we hope that we were able to correct all mistakes. 

2. Second: There is no research question. Normally, the literature is reviewed, an issue arising from this review is presented, and this issue is then condensed into a scientific question of importance (at least for the field), and then addressed with an experimental study aimed at answering that specific question as well as possible. The implicit research question here seems to be "does this new task we invented, which we think might measure cognitive flexibility, and which we have reason to believe is more social and natural than the traditional ones, elicit the effects which we think it should?" This is a very specific and ad-hoc question, and the authors do not make clear how this specific issue is of general interest for the field of cognitive psychology.

We agree with the reviewer that the former version of our manuscript missed the opportunity to clearly describe our research question. As we now state in the first paragraph of our introduction, we here ask which possible mechanisms are involved in social flexibility, i.e. our ability to adapt to unexpected events in social interaction (page 3). This research question is derived from the theoretical considerations of cognitive flexibility in non-social settings that has been the focus of research of classical cognitive research for decades. 

3. Third, there is no good theoretical motivation for the explored task. The authors claim that it is more natural and more social, because it involves one person's utterance to be the input for the next person's task, and so on. This of course creates interdependency between what the two participants do, but that does not necessarily make the task natural or social. The ecological validity of the task is, I believe, very low, and if the authors would want to claim that it is sufficiently high, they need to provide clear arguments for that claim. Personally, I find it hard to imagine many situations situation in our daily lives where the ability to collectively formulate a grammatical sentence by alternatingly and spontaneously providing a word would be useful. Also, the task seems fundamentally different in the level of processing from the three other tasks that it is compared with. Contrary to the other tasks, which all involve cognitive control at relatively peripheral levels, the new task involves a number of simultaneous higher-cognitive subtasks: the parsing of the words, the constructing of higher-level linguistic representations, word recognition, semantic processing, and memory components. Each of these sub-tasks literally have entire scientific sub-fields devoted to studying them. And these sub-processes also run in parallel in the participant's brain, which adds an entire additional level of complexity. This means that there are so many things happening in the participant's mind/brain during this task that whatever the results that were to come out of it, they are very difficult to be pinned down to specific cognitive variables. And I see no compelling reason to believe that the task is more sensitive to cognitive flexibility than to any of the other variables that are involved in the list of simultaneously running sub-processes in my, almost certainly incomplete, list above.

We thank reviewer for these valuable comments. We agree with the reviewer that the Word-by-Word task requires a variety of different cognitive processes such as word-recognition or memory components that we did not address in our theoretical consideration or data analyses. We added these considerations in an additional limitations section “a critical evaluation of the Word-by-Word task” in the discussion (page 21-22). However, we argue that this is not fundamentally different in comparison to classical cognitive task. “Simple” cognitive tasks that measure cognitive flexibility (such as the Simon task or task-switching paradigms) also include a variety of different sub-processes that are necessary to execute the tasks – but not targeted by the research question they aim to answer. For example, classical cognitive tasks also involve stimuli-recognition or memory effects. This is why we (and cognitive psychologists usually) perform with-subjects measures that use the same task with the same set of many processes involved across conditions and varied properties between conditions that should influences the processes of interest. 

However, we agree that the manuscript needed to better define these process of interest. In the revised version of the manuscript we provide a model that describes which processes are possibly involved in the Word-by-Word task. With that we do not claim that we describe ALL underlying processes, we simple point to the ones in our interest and which could be mapped to similar process in the classical cognitive tasks. 

We used the Word-by-Word task also in former research in different versions with different levels of experimental control. We found reliable behavioral and neural findings that clearly suggest that participants have more difficulties in handling unexpected events in all three versions of the Word-by-Word task:

Goregliad Fjaellingsdal, T., Schwenke, D., Ruigendijk, E., Scherbaum, S., & Bleichner, M. G. (2020). Studying brain activity during word-by-word interactions using wireless EEG. PLoS ONE, 15(3), 1–21. 

Goregliad Fjaellingsdal, T., Schwenke, D., Scherbaum, S., Kuhlen, A. K., Bögels, S., Meekes, J., & Bleichner, M. G. (2020). Expectancy effects in the EEG during joint and spontaneous word-by-word sentence production in German. Scientific Reports, 10(1), 5460. 

We can hence conclude that this task could serve as a valid tool for an investigation of cognitive processes that also include interactive aspects of human cognition. 

4. Finally, the results of the study are not encouraging and hard to interpret. First, there was no significant effect on visibility. The authors suggest that that was because the participants may not have chosen to exploit the visual channel, but another possibility is that the study was just severely underpowered, as the power analysis on which the design was based was on a within-participant effect of medium strength. Not on a potentially small effect in a between-participant effect (which has much lower power). Second, the TTRT was not correlated with any of the standard flexibility tasks. Third, the CE difference was only weakly correlated with one of the tasks, while the p-value of that test was not corrected for the number of correlation tests (e.g. using the Bonferroni method). So according to the results, the task seems not to behave as expected or desired. This is, however, not the conclusion that the authors draw.

We thank reviewer for pointing us toward these important aspects. First, we agree with the reviewer that our sample size might be too small for a between-subject-comparison. We added this limitation in our discussion section (page 21). We also added that the correlative findings should be considered with caution since the correlation was rather small and only with the Stop-Signal task page 20). However, we believe that in this case it is not necessary to use Bonferoni corrections for the following reason: Even though we first refrained from including the model that is now included in the reworked manuscript, out hypotheses were based on this theoretical foundation that allows a direct link between the Word-by-Word task and classical cognitive measures. To some extent the analyses are still explorative since we measured three classical tasks without any specific prediction. However, as the reviewer pointed out him or herself, the sample size in our research is too small to allow any further correction. We added this limitation to the discussion and argue that the Word-by-Word task still serves as a valid and reliable measure to address response conflict within the participants. We found such effects not only in this research but also in our former research with a comparable Word-by-Word design (see point 3).

In closing, we thank the reviewers for their very helpful and constructive comments and we thank you again for your editorial efforts on this paper. We hope that we addressed sufficiently all of yours and the reviewers’ concerns and are looking forward to your decision.

Sincerely,

Diana Schwenke

---

## [Editor Report · Decision Letter 1]

9 Jun 2020

An approach to social flexibility: Congruency effects during spontaneous word-by-word interaction

PONE-D-20-00636R1

Dear Dr. Schwenke,

We’re pleased to inform you that your manuscript has been judged scientifically suitable for publication and will be formally accepted for publication once it meets all outstanding technical requirements.

Kind regards,

Marte Otten, Ph.D.

Academic Editor

PLOS ONE

---

## [Editor Report · Acceptance letter]

11 Jun 2020

PONE-D-20-00636R1 

An approach to social flexibility: Congruency effects during spontaneous word-by-word interaction 

Dear Dr. Schwenke:

I'm pleased to inform you that your manuscript has been deemed suitable for publication in PLOS ONE. Congratulations! Your manuscript is now with our production department. 

Kind regards, 

on behalf of

Dr. Marte Otten 

Academic Editor

PLOS ONE